# Radiosensitizing Effect of PARP Inhibition on Chondrosarcoma and Chondrocyte Cells Is Dependent on Radiation LET

**DOI:** 10.3390/biom14091071

**Published:** 2024-08-27

**Authors:** Antoine Gilbert, Mihaela Tudor, Amandine Delaunay, Raphaël Leman, Julien Levilly, Alexandre Atkinson, Laurent Castéra, Anca Dinischiotu, Diana Iulia Savu, Samuel Valable, François Chevalier

**Affiliations:** 1UMR6252 CIMAP, CEA-CNRS-ENSICAEN, Normandie Université, Team Applications in Radiobiology with Accelerated Ions, 14000 Caen, Franceamanddelaunay@gmail.com (A.D.); 2Department of Life and Environmental Physics, Horia Hulubei National Institute for R&D in Physics and Nuclear Engineering, Reactorului 30, 077125 Magurele, Romania; mihaela.tudor@nipne.ro (M.T.);; 3Faculty of Biology, University of Bucharest, Splaiul Independentei 91-95, 050095 Bucharest, Romania; 4Laboratoire de Biologie et de Génétique du Cancer, Centre François Baclesse, 14000 Caen, France; r.leman@baclesse.unicancer.fr (R.L.); a.atkinson@baclesse.unicancer.fr (A.A.); l.castera@baclesse.unicancer.fr (L.C.); 5Inserm U1245, Cancer Brain and Genome, Normandie Universite, UNICAEN, FHU G4 Genomique, 76000 Rouen, France; 6Université de Caen Normandie, CNRS, Normandie Université, ISTCT UMR6030, GIP CYCERON, 14000 Caen, France

**Keywords:** carbon ion irradiation, chondrosarcoma, PARP inhibitor, DNA damages

## Abstract

Chondrosarcoma is a rare malignant tumor that forms in bone and cartilage. The primary treatment involves surgical removal of the tumor with a margin of healthy tissue. Especially if complete surgical removal is not possible, radiation therapy and chemotherapy are used in conjunction with surgery, but with a generally low efficiency. Ongoing researches are focused on understanding the genetic and molecular basis of chondrosarcoma following high linear energy transfer (LET) irradiation, which may lead to treatments that are more effective. The goal of this study is to evaluate the differential effects of DNA damage repair inhibitors and high LET irradiation on chondrosarcoma versus chondrocyte cells and the LET-dependency of the effects. Two chondrosarcoma cell lines with different *IDH* mutation status and one chondrocyte cell line were exposed to low LET (X-ray) and high LET (carbon ion) irradiation in combination with an Olaparib PARP inhibitor. Cell survival and DNA repair mechanisms were investigated. High LET irradiation drastically reduced cell survival, with a biological efficiency three times that of low LET. Olaparib significantly inhibited PARylation in all the tested cells. A significant reduction in cell survival of both chondrosarcoma and chondrocyte cells was observed following the treatment combining Olaparib and X-ray. PARP inhibition induced an increase in PARP-1 expression and a reduced effect on the cell survival of WT *IDH* chondrosarcoma cells. No radiosensitizing effect was observed in cells exposed to Olaparib paired with high LET irradiation. NHEJ was activated in response to high LET irradiation, neutralizing the PARP inhibition effect in both chondrosarcoma cell lines. When high LET irradiation is not available, PARP inhibition could be used in combination with low LET irradiation, with significant radiosensitizing effects on chondrosarcoma cells. Chondrocytes may be affected by the treatment combination too, showing the need to preserve normal tissues from radiation fields when this kind of treatment is suggested.

## 1. Introduction

Chondrosarcomas are the second most frequent malignant bone tumors (20–30% of bone sarcomas). They are well known for their resistance to chemotherapy and conventional X-ray radiotherapy [1,2]. Indeed, the first treatment indicated is a curative surgery. However, this treatment cannot be applied in all cases, such as advanced stage and intricately located tumor (i.e., skull base chondrosarcoma). New therapy strategies are needed in these situations [3,4].

Recently, the use of hadrontherapy appears to be promising for the treatment of these tumors [5]. Hadrons present the advantage of depositing their energy in the form of a peak called the Bragg peak, thus allowing better control of the dose deposited in the tumor and limiting the irradiation to the surrounding healthy tissue. In addition, for heavier accelerated ions such as carbon ions (C-ions), a higher LET makes it possible to increase damage at the cellular level, in particular by creating more complex DNA damage, which is more difficult to repair [5]. Hadrontherapy has demonstrated its effectiveness in the treatment of sarcomas in pre-clinical and clinical studies [6,7,8,9]. Furthermore, the development of hadrontherapy centers around the world offers real hope for the treatment of these tumors.

Concomitant treatments consisting of using targeted pharmaceutical agents such as PARP inhibitors can improve the effect of irradiation [10,11,12,13,14,15]. PARP proteins are mainly involved in base excision repair (BER), a DNA single-strand damage repair pathway [16]. These agents were first used to achieve synthetic lethality in the case of BRCA mutated tumors, with cells deficient in Homologous recombination (HR), a DNA double-stranded damage repair [17]. Moreover, pre-clinical studies showed that PARP inhibitors (PARPi) like Olaparib and talazoparib radiosensitized chondrosarcoma cells against both conventional and hadron radiation [10,13]. Interestingly, it was observed that the radiosensitizing effect induced by PARPi was dependent on the radiation’s quality.

It is well known that treatment procedures can be personalized and guided by the mutation status of cancer cells. IDH1/2 mutations are frequently found in chondrosarcomas. Some studies showed that PARPi are effective for the treatment of gliomas and leukemias mutated on IDH1/2 [18,19]. Concerning chondrosarcoma, a clinical study showed that Olaparib was efficient for the treatment of IDH1/2 mutated chondrosarcomas [20]. On the other hand, in vitro studies on several chondrosarcoma cell lines demonstrated that the effect of talazoparib is not dependent on the IDH1/2 status [21,22].

This study aims to investigate the potential of Olaparib to radiosensitize chondrosarcoma and chondrocyte cells with differential IDH statuses to C-ions and X-ray. We observed a differential response of the tested cell lines. The response depended on the IDH status and LET of radiation. Several molecular mechanisms are proposed to explain the level of radiosensitization of the cells.

## 2. Materials and Methods

### 2.1. Cell Cultures

The chondrosarcoma OUMS27 cell line was initiated from a primary grade III chondrosarcoma located on the left humerus from a 65-year-old male [23]. The JJ012 cell line was initiated from a primary grade II chondrosarcoma from a 39-year-old patient [24] and kindly provided by Dr. J. A. Block (Rush University medical center, Chicago, IL, USA). The chondrocyte cell line MC615 was initiated from primary embryonic mouse limb chondrocytes [25]. This cell line was used as a chondrocyte model. According to previous studies, cell line MC615 displays a chondrocyte phenotype and metabolism [26,27,28]. The three cell lines used in this study were grown in RPMI 1640 (Merck, Darmstadt, Germany) supplemented with 10% fetal calf serum (Merck, Darmstadt, Germany), 1% penicilline-streptomycine (Merck, Darmstadt, Germany), and 2 mM L-glutamine (Merck, Darmstadt, Germany), at 37 °C in humidified atmosphere with 5% CO_2_.

### 2.2. Drug Treatments

Olaparib (AZD-2281; AstraZeneca, Cambridge, UK) was purchased from Cell Signaling Technology (Danvers, MA, USA). First, Olaparib powder was dissolved in DMSO at 10 mM to create a stock solution. Concentrations from 0.5 to 40 µM were used to assess viability, and a single dose of 2 µM was selected for further experiments. Before irradiation, cells were treated with the drug and incubated for 2 h in standard conditions. Negative control samples were treated with the same volume of DMSO as the test samples (0.02%, *v*/*v*).

### 2.3. Irradiation

X-rays were delivered by a Faxitron (CellRad, Precision X-ray, Madison, CT, USA) using a copper filter, with a tube tension of 129 keV and an intensity of 4.7 mA (corresponding to a dose rate of 1 Gy/min). Cells were irradiated with doses between 1 and 6 Gy.

For carbon ion irradiation, experiments were performed at GANIL (Caen, France) using the IRABAT beam line with a native carbon ion beam of 95 Mev/u [29]. A PMMA degrader 16.9 mm in thickness was used to obtain a LET of 73 keV/µm (2 Gy = 1.71 × 10^7^ particles/cm^2^). Cells were exposed to doses between 1 and 3 Gy of C-ion radiation. Cells were irradiated in 25 cm^2^ flasks, at 80% of confluency.

### 2.4. Clonogenic Assay

Cells were harvested 24 h after irradiation, counted with trypan blue (results in Appendix A) and re-plated in appropriate densities in 6-well plates. After 7–12 days (depending on the cell line), the colonies were stained with a Crystal Violet solution (0.3% *w*/*v* Crystal Violet, Sigma-Aldrich (Merck, Darmstadt, Germany), in 20% *v*/*v* ethanol). Colonies with more than 50 cells were counted under a stereomicroscope. Results were expressed as a percentage of non-irradiated cells. The CS-cal program [30] was utilized to model survival curves. The X-ray data were fitted according to a linear quadratic model, while the carbon ion experiments were fitted using a linear model as previously described [31].

### 2.5. MTT Toxicity Test

Cells were seeded in 96-well plates. Next, 24 h after seeding, the cells were incubated with different concentrations of Olaparib (0.5, 1, 2, 5, 10, 20, 40 µM) for 48 h. Then, the cells were washed and an MTT (3-(4,5-dimethyl-2-thiazolyl)-2,5-diphenyl-2-*H*-tetrazolium bromide) solution (0.5 mg/mL, diluted in medium without FBS) was added, followed by a 3 h incubation period. The formazan blue crystals were then solubilized with a 0.04 M HCL/isopropanol solution. The optical density was read by a microplate reader at a 595 nm wavelength (Mutiskan FC, Thermo Scientific, Waltham, MA, USA). Results were expressed as the mean of absorbance normalized to the control by 3 independent experiments.

### 2.6. Sequencing

We started the sequencing from 100 to 200 ng of DNA extract using either a Maxwell automate or a Qiagen kit (QIAamp DNA FFPE Tissue, catalog No. 56404, Hilden, Germany). The DNA was fragmented using a Covaris E220 (Covaris, Woburn, MA, USA). The fragmented DNA was processed with a SureSelect XT HS2 (catalog No. G9983A; Agilent, Santa Clara, CA USA). A custom panel of 127 genes was considered for the enrichment of regions of interest. Our panel included the 15 HRR genes (*BRCA1*, *BRCA2*, *ATM*, *BARD1*, *BRIP1*, *CDK12*, *CHEK1*, *CHEK2*, *FANCL*, *PALB2*, *PPP2R2A*, *RAD51B*, *RAD51C*, *RAD51D*, *RAD54L*). After normalization, the libraries were sequenced on a NextSeq 500 device (Illumina, San Diego, CA, USA). The fastq files were generated from the raw data of sequencing by Bcl2Fastq v2.20 (RRID:SCR_015058). Alignment was performed using BWA v0.7.17 (RRID:SCR_010910), using the genome assembly version GRCh37/hg19. PCR duplicates were removed by Picard (RRID:SCR_006525) v2.21.7. The identification of genomic events, such as copy number variations, was done by CNVkit pipeline v0.9.7 (RRID:SCR_021917). The calculation of a genomic instability score was done using the GIScar test [32]. Three variant callers were used to detect the single nucleotide variants and small indels from the aligned data: Haplotype Caller embedded in GATKv4.1.6.0 (RRID:SCR_001876), LoFreq (RRID:SCR_013054), and OutLyzer [33]. The variants were annotated by the ANNOVAR tool (RRID:SCR_012821). The variants were interpreted according to the guidelines provided by the French Genetics and Cancer Group [34]. Only pathogenic or likely pathogenic variants were reported.

### 2.7. Western Blotting Analysis

Cells were collected 24 h after irradiation, washed with PBS, and stored at −80 °C to be further used for protein extraction. Cell pellets were lysed with a lysis buffer containing 25 mM of Tris-Base, 120 mM of NaCl, 10 mM of Triton X and 1 mM of EDTA. The lysis buffer was supplemented by a cocktail of 1 mM of PMSF, protease and phosphatase inhibitors (Merck, Darmstadt, Germany). Cell extracts were denaturated using Laemlli 4× (Bio-Rad, Marnes-la-Coquette, France) and β-mercaptoethanol at 95 °C for 5 min. The extracted samples were then separated by SDS-PAGE on a TGX 4–15% gradient polyacrylamide gel (Bio-Rad, Marnes-la-Coquette, France) and transferred to a polyvinylidene difluoride (PVDF) membrane. Membranes were blocked by Azure Fluorescent Blot blocking buffer (Azure Biosystems, Dublin, OH, USA) for 1 h. Membranes were incubated with anti-Poly-ADP-ribose (1:1000, MABC547 Merck, Damstadt, Germany), anti-Poly-ADP-Ribose-Protein (1:1000, #90959, Merck, Darmstadt, Germany) anti-p.ATM (S1981) (D25E5) (1:1000, #13050, Cell Signaling, Danvers, MA, USA), anti-p.BRCA1 (1:1000, S1524) (#9009, Cell Signaling, Danvers, MA, USA), anti-p. DNAPK (S2056) (1:1000, AB124918, Abcam, Cambridge, UK), or anti-beta-actin (1:5000, A1978, Merck, Darmstadt, Germany) diluted in TBS-T with 1% skim milk powder overnight at 4 °C. Then, the blots were washed 3 times with TBS-T and then incubated with goat-anti-Rabbit Azure Spectra 700 (1:10,000, Azure Biosystems, Dublin, OH, USA) or with goat-anti-Mouse Azure Spectra 800 (1:10,000, Azure Biosystems, Dublin, OH, USA) diluted in TBS-T with 1% skim milk powder for 1 h. Blots were washed 3 times with TBS-T before reading their infrared signals with Ultimate Western Blot Imager Azure 600 (Azure Biosystems, Dublin, OH, USA). Signal quantifications were performed using Image Studio Lite 5.0 software. Results are expressed as a semi-quantification regarding β-actin, normalized with the non-irradiated condition, without drugs, for three independent experiments. Quantification is associated with cropped images; the cropping focuses attention on specific signals. Uncropped images are included in Appendix A.

### 2.8. Statistics

The statistical analysis was performed using the statistical module in GraphPad Prism 8. For multiple comparisons (the clonogenic assay, MTT assay and western blot DNA repair study), an Analysis of Variance (ANOVA) followed by a Tukey’s multiple comparison was applied. For the comparison of the two populations (Olaparib toxicity on plating and WB), a *t*-test was used. In both cases, data were considered to be significantly different when *p* < 0.05 (*).

## 3. Results

### 3.1. Clonogenic Survival of Chondrosarcoma and Chondrocyte Cells Is Dependent on LET

Without irradiation, the plating efficiency of OUMS27, JJ012 and MC615 cells was 45% ± 4%; 30% ± 3% and 49% ± 6% (at least *n* = 3), respectively. The clonogenic survival of both chondrosarcoma cell types (JJ012 and OUMS27) and chondrocytes (MC615) was evaluated following X-ray and C-ion irradiation.

In case of X-ray irradiation, very small differences were observed between the three cell lines analyzed (Figure 1, blue lines). The most resistant cell line for each X-ray dose was MC615 (blue circles), with a D10 of 7.58, whereas the most sensitive cell line was OUMS27 (blue squares), which displayed a significant difference at 1 Gy. The D10 of the JJ012 and OUMS27 chondrosarcoma cell lines was 7.5 and 6.17, respectively (Table 1).

Following C-ion irradiation, as expected, the D10 was significantly reduced for each cell line. Similarly, as in the case of X-ray irradiation, MC615 cells were the most resistant, while OUMS27 cells were the most sensitive. The D10s of the MC615, JJ012 and OUMS27 cells were 2.27, 2.24 and 2.02, respectively, (Table 1) with significant differences at doses of 1 and 2 Gy between the MC615 and OUMS27 cells. The relative biological effectiveness (RBE) of C-ions has very similar values between cell lines, measuring 3.37 (JJ012), 3.21 (MC615) and 3.05 (OUMS27), in agreement with the commonly observed RBE values for C-ions.

### 3.2. Similar Low Toxicity and Effective Activity of PARP Inhibitor

Toxicity tests of Olaparib molecules were performed using an MTT test on each cell line at different concentrations ranging from 0.5 to 40 µM of Olaparib, using a control sample with only DMSO as a comparison (Figure 2A). For the next experiments, the dose of 2 µM was selected and used for all cell lines. Indeed, 2 µM of Olaparib induced a low (<10%) and acceptable level of toxicity with OUMS27 cells (2.5% toxicity +/− 0.2%), JJ012 cells (7.5% toxicity +/− 0.7%) and MC615 cells (10% toxicity +/− 1%). This low toxicity was also observed with trypan blue staining before clonogenic assay (Appendix A).

The plating efficiency of each cell line was evaluated after the treatment with 2 µM of Olaparib (Figure 2B). No significant effect was observed between the control and PARPi treated cells.

Then, the PARylation activity of the cells was checked and proved by Western blotting using an antibody specific to this post-translational modification (Figure 2C). Olaparib at 2 µM successfully inhibited protein PARylation for all cell lines.

### 3.3. Differential Mutation Status and Genomic Instability: GIScar Score

The mutation status of cancer cells is the main criterion of the use of PARPi in the clinic. Homologous recombination defective (HRD) cells are more sensitive to PARPi, allowing a synthetic lethality during DNA damage repair. Indeed, PARPi mainly blocks the repair of DNA single-strand break (SSB) damages by inhibiting the base excision repair (BER) pathway. Therefore, an accumulation of SSB leads to double strand breaks (DSB) during DNA replication. HRD cells with toxic accumulations of DSB are unable to repair DNA damage properly, leading to DNA mutations caused by non-homologous end joining NHEJ repair, cell cycle blockages and cell death.

A deep mutation analysis was performed on each chondrosarcoma cell line (Table 2). OUMS27 and JJ012 cells carry a TP53 gene mutation. Several genes were mutated with a missense substitution in JJ012 cells (IDH1, SLX4, ARID1B) or with a frameshift (ARID1A). In the case of OUMS27 cells, we identified only an intronic mutation in the EMSY gene. According to this analysis, no clear impact of mutation status on homologous recombination (HR) could be established.

An analysis of copy number variations (CNV) was done on each cell line, and a GIScar score was determined (Table 2). The data showed that the OUMS27 cells presented a higher genomic instability level than the JJ012 cells. We detected 27 large genomic events in OUM27 cells instead of 6 in JJO12 cells and the GIScar scores were 0.408 and 0.154, respectively. The scatter plot of the CNV kit tool also illustrated this difference in genomic instability levels (Figure 3). However, to our knowledge there is not a clear definition of the decision-making threshold for classifying chondrosarcoma cell lines as HR-proficient or HR-deficient.

### 3.4. Radiosensitizing Effect of PARPi Depends on Radiation LET

Following the cellular characterization of the cells and the toxicity of PARPi, we analyzed the cumulative effects of irradiation and PARPi on each cell line (Figure 4).

In combination with X-ray irradiation, PARPi induced a significant reduction in the clonogenic survival of all cells (Figure 4A–C blue dotted lines). Olaparib had the highest radiosensitizing effect for the JJ012 cell line (Table 1), which displayed the highest enhancement ratio (ER = 1.5). This ratio was lower in the OUMS27 cells (ER = 1.19) and intermediate in the MC615 cells (ER = 1.37). This effect of PARPi in combination with X-ray irradiation was observed in the initially seeded cells, 24 h after irradiation, when the cells were counted with trypan blue (Appendix A).

Conversely, regardless of the dose applied, no radiosensitizing effect associated with PARPi was seen in relation to C-ion irradiation in any of the cell lines (Figure 4A–C red dotted lines). A significant effect of PARPi on the JJ012 cells’ viability was observed 24 h after carbon ion irradiation (Appendix A), with no impact on colony forming. This effect was certainly due to an additive effect of irradiation and PARPi on these cells at 24 h, but with no final impact on cell clonogenic survival.

### 3.5. Specific Activation of DNA Repair Pathways Depends on Radiation LET

To investigate the ability of irradiation associated with PARP inhibition to alter DNA repair, whole cell lysates of OUMS27 and JJ012 chondrosarcoma cells were harvested 24 h following 4 Gy X-ray or 2 Gy C-ion irradiation and prepared for western blot analysis (Figure 5). Quantifications are presented for one representative blot, and beta-actin was used as a loading control (mean, +/−SD; *n* = 3).

The expression of proteins associated with BER (PARP-1), HR (p-ATM and p-BRCA-1) and NHEJ (p-DNA-PKc) were quantified with (+) or without (−) a PARP inhibitor. In OUMS27 cells, PARPi induced a significant increase in PARP-1 expression with and without irradiation. This increase was even more pronounced with C-ion irradiation (x4), as compared with X-ray (x2). In contrast, no modification of PARP-1 expression was observed in JJ012 cells, whatever the treatment condition, +/− PARPi and +/− irradiations. In non-irradiated samples, PARPi did not activate ATM, DNA-PKc and BRCA-1 phosphorylation. A significant increase in the p-DNA-PK level was observed following irradiation in both cell lines. This activation of NHEJ was higher with PARPi associated with C-ions in the case of OUMS27 cells. A similar activation is observed with p-ATM and p-BRCA-1 markers following irradiation, and in association with PARPi, for the OUMS cell line. In the case of JJ012 cells, a lower activation seemed to occur, with a slightly increased level of p-ATM (*p* > 0.05) after X-ray, but with a significant increase in p-BRCA-1 (*p* < 0.05) following C-ions irradiation in combination with PARPi.

## 4. Discussion

Chondrosarcoma is a rare type of cancer that originates in the cartilage cells, primarily affecting the bones. Unlike other more common types of bone cancer, chondrosarcomas are notorious for their resistance to standard chemotherapy and radiotherapy treatments [4]. This resistance significantly limits therapeutic options, posing a substantial challenge for oncologists and impacting patient prognosis. While surgery remains the cornerstone of therapy, ongoing research into targeted therapies, combination treatments, and advanced radiotherapy techniques offers hope for improved outcomes [35].

Several mechanisms are proposed to explain chondrosarcoma’s resistance to treatments. Chondrosarcomas often exist in hypoxic environments, which can induce the expression of genes that protect cancer cells from the effects of chemotherapy and radiotherapy [35,36,37]. The dense and rigid extra-cellular matrix in chondrosarcomas acts as a physical barrier, preventing effective drug penetration and radiation absorption [23]. In addition, specific genetic mutations and epigenetic changes in chondrosarcoma cells can contribute to resistance by altering cell cycle regulation, apoptosis pathways, and metabolic processes. Finally, enhanced DNA repair capabilities in chondrosarcoma cells can mitigate the DNA damage induced by chemotherapy and radiotherapy [38].

In the present study, we proposed to combine PARPi with high LET irradiation in the treatment of chondrosarcoma. The original goal was to further enhance DNA damage. Indeed, high LET radiation induces more complex DNA damage compared to low LET radiation [39]. PARPi prevent the repair of single-strand DNA breaks, which when combined with high LET radiation-induced DNA damage, can lead to synergistic effects, causing cell death in chondrosarcoma cells [11]. Chondrosarcoma cells often have defects in their DNA repair pathways. PARPi exploit this vulnerability by inhibiting an alternative DNA repair pathway, leading to an accumulation of DNA damage. Combining them with high LET irradiation further overwhelms the cells’ ability to repair DNA damage, resulting in increased tumor cell death, as observed with the CH2879 cell line [13]. In HeLa cells, PARP-1 inhibition in combination with carbon ion irradiation significantly reduced MMPs’ activity, showing potential improvement of carbon ion therapy with PARP-1 inhibition to control metastatic processes [40].

In this study, we observed an RBE of about 3 independently on the cell lines investigated when C-ions were compared with X-ray (Table 1; Figure 1). Both chondrosarcoma and chondrocyte cells were more sensitive to C-ions. This outcome clearly shows the need to properly preserve healthy tissue around the tumor, in order to limit the side effects and the toxicity of hadrontherapy.

Based on our experiments, we selected a concentration of Olaparib (2 µM) which significantly inhibited PARylation formation by PARP proteins (Figure 2C,D) and had no cellular toxicity (Figure 2B). In association with X-ray irradiation, we observed that Olaparib can increase the effect of irradiation alone in MC615 cells and JJ012 cells (IDH mutated), with an enhancement ratio of about 1.4 and 1.5, respectively. In contrast, OUMS cells (IDH WT) exhibited a lighter response and only at the highest dose. According to these results, IDH mutation seemed to be a good marker for the efficiency of PARPi on the two chondrosarcoma cells line tested. IDH mutation was previously shown to impact the immune microenvironment in tumors treated with PARPi and radiotherapy, which also increased PD-L1 expression and enhanced checkpoint inhibition [41].

Again, in association with X-ray irradiation, Olaparib could increase the side effects on normal tissues, showing the need to focus irradiation only on the tumor volume. This is one of the characteristics of accelerated ions. C-ions present a better ballistic track when compared with X-ray, according to a reduced lateral scattering and longitudinal straggling compared to low LET radiations such as protons and X-rays [42]. In addition, the dose deposit at the end of the track, according to the Bragg peak, improves the dose-depth distribution in radiotherapy and reduces side effects on surrounding normal tissues. In the present study, we observed that Olaparib did not improve the effect of carbon ions alone (Figure 4). Indeed, an enhancement ratio close to 1 was observed with all cells (Table 1). We sought to understand the mechanism of action of Olaparib on the chondrosarcoma cell lines. Upon analyzing the mutation status of these cells, we identified several mutated genes that are involved in DNA repair. The GIScar score allowed us to evaluate the response of these cells to PARPi, with a higher score in OUMS27 cells (0.4), as compared with JJ012 cells (0.15). This score was not in agreement with the enhancement ratio of Olaparib observed with X-ray. JJ012 cells were more affected by Olaparib, as compared with OUMS27 cells, demonstrating the complexity of PARP inhibition, which cannot be related only to specific gene mutations. PARPi were initially used on BRCA mutated cancers, but BRCA proficient cells could also be sensitized by PARPi when associated with X-rays/protons/C-ions, showing the high therapeutic potential of these inhibitors [43].

Then, we analyzed several proteins involved in DNA repair, and we observed a specific and contrasted response to PARP’s inhibition of the two chondrosarcoma cell lines. Indeed, in response to PARPi, a significant increase in PARP-1 protein levels was observed in OUMS27 cells. There was no modification of expression in JJ012 cells. This specific response of OUMS27 cells could explain the resistance to PARPi; an increased level of PARP-1 could counteract the action of PARPi in this cell line, even if the PARylation is reduced at the same time. PARP-1 is a major actor of DNA repair, involved not only in BER responses but also in HR and NHEJ, at different levels [44]. Moreover, since no PARP-1 cleavage could be observed in both cell lines, no apoptosis occurred following the PARP inhibition associated with irradiation. In addition, we observed a general response of both cell lines to irradiation, with a specific phosphorylation of ATM and BRCA-1 after 4 Gy of X-ray and 2 Gy of C-ions, especially with PARP inhibition. This proved the involvement of both homologous recombination and non-homologous end joining in DNA damage responses as observed at the level of the phosphorylation of DNA-PKc.

It is interesting to note that PARP inhibition increased DNA-PKc phosphorylation in OUMS27 cells when combined with carbon ion irradiation, but had no effect on clonogenic survival (ER = 1). In this condition, NHEJ activation in OUMS27 cells allowed an efficient DNA repair, which neutralized the effect of the PARP inhibition. A similar effect was observed in JJ012 cells, with a high increase of NHEJ activation following carbon ion irradiation with and without PARP inhibition.

## 5. Conclusions

This study highlighted the complexity of DNA repair mechanisms and cell responses to DNA damages, depending on LET and the type of DNA repair inhibitors. PARP inhibition showed promising effects in association with X-ray irradiations in both chondrosarcoma cell lines, but to a higher extent in IDH mutant cells. In non-IDH-mutated cells, PARP-1 expression was increased in response to PARP inhibition, which could explain the reduced radiosensitizing effect in combination with X-ray. No sensitizing effect was observed when PARP inhibition was combined with high LET irradiation, particularly through the NHEJ activation of chondrosarcoma cells. Normal chondrocytes responded similarly to PARP inhibition and low LET irradiation and demonstrated the significant radiosensitizing effect of Olaparib. Since this molecule is administered in a systemic way, such sensitizing effects could occur within irradiated normal tissues or around the irradiation field of a tumor, inducing sequelae and adverse outcomes.

## Figures and Tables

**Figure 1 biomolecules-14-01071-f001:**
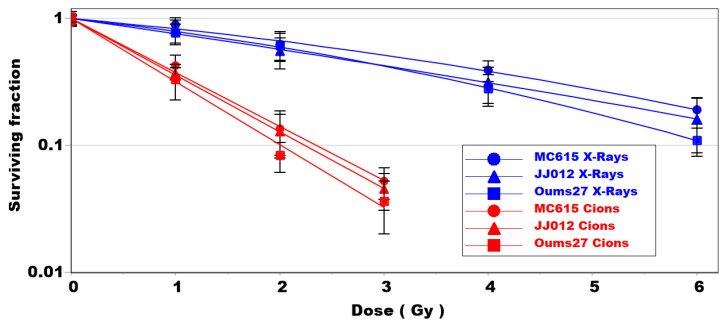
Clonogenic survival of MC615 cells (circle), JJ012 cells (triangle) and OUMS27 cells (square) after X-ray (blue lines) and C-ions (red lines) irradiation. (3 independent experiments).

**Figure 2 biomolecules-14-01071-f002:**
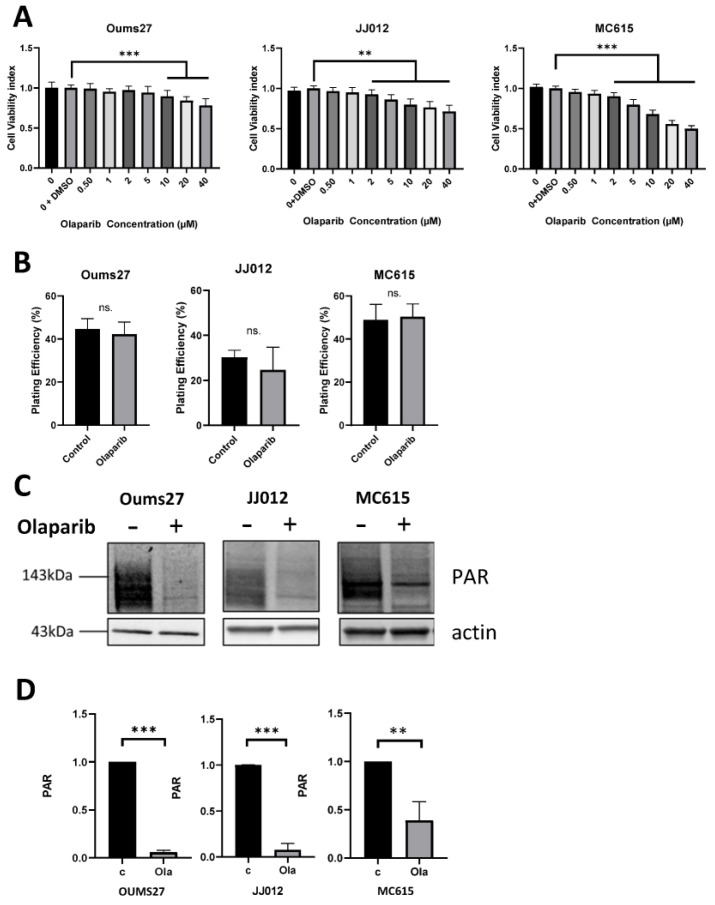
(**A**) Cell viability index (%) using the MTT cytotoxicity test on MC615, JJ012 and OUMS27 cells with increased concentration of Olaparib from 0 to 40 µM (**: *p* < 0.01, ***: *p* < 0.001, with three independent experiments); (**B**) Plating efficiency (%) of MC615, JJ012 and OUMS27 cells with or without (control) Olaparib at 2 µM (ns. non-significant differences with three independent experiments); (**C**) western blotting with PARylation signal (PAR) of MC615, JJ012 and OUMS27 cells with (+) and without (−) Olaparib, actine is used as loading control; (**D**) Quantification of the Western blotting signals from “part C” with at least 3 independent experiments (**: *p* < 0.01, ***: *p* < 0.001). Original images can be found in Appendix A.

**Figure 3 biomolecules-14-01071-f003:**
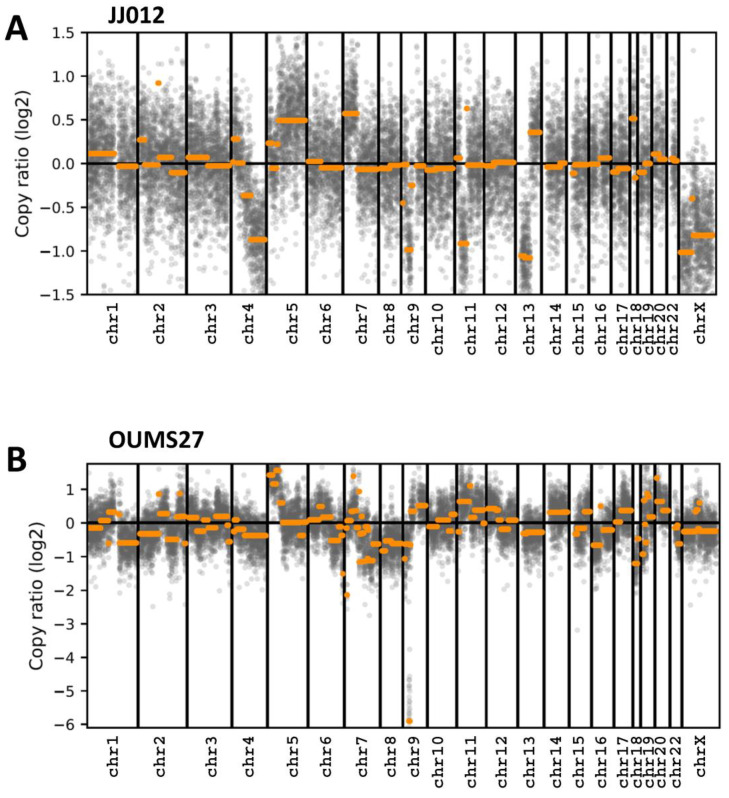
Copy number variation (CNV) analysis of JJ012 cells (**A**) and OUMS27 cells (**B**).

**Figure 4 biomolecules-14-01071-f004:**
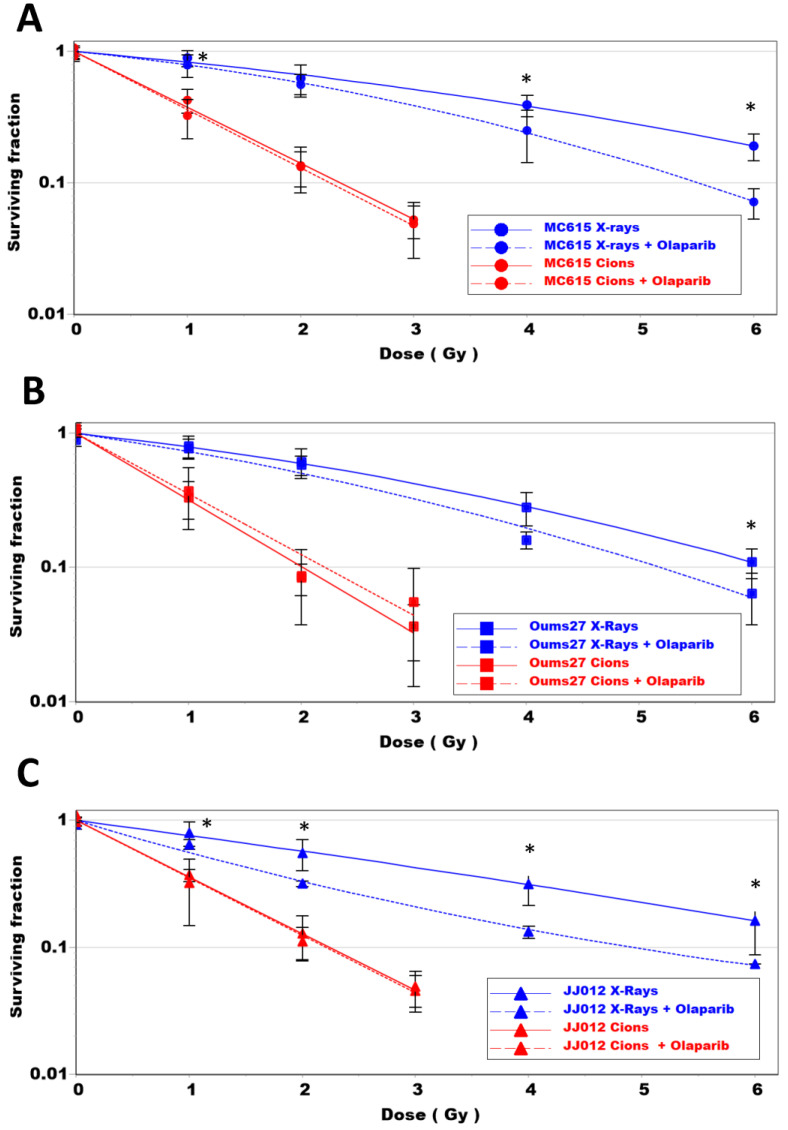
Clonogenic survival of MC615 cells (**A**), OUMS27 cells (**B**) and JJ012 cells (**C**) after X-ray (blue lines) and C-ions (red lines) irradiation, with (dotted line) and without (full lines) Olaparib. Mean of at least 3 independent experiments (*: *p* < 0.05).

**Figure 5 biomolecules-14-01071-f005:**
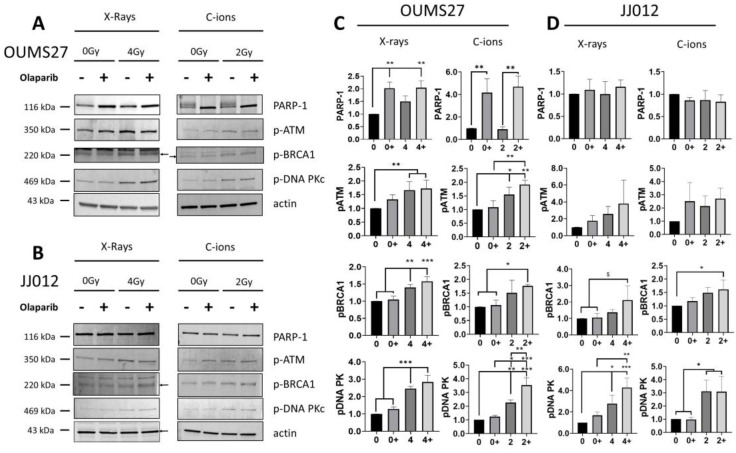
Western blotting analysis of OUMS27 (**A**) and JJ012 (**B**) cells after X-ray and C-ions irradiation with (+) and without (−) Olaparib PARPi. Antibodies against PARP-1, phospho-ATM (p-ATM), phospho-BRCA1 (p-BRCA1) and phospho-DNA-PKc (p-DNA-PKc) were used for each conditions, beta-actin is used as loading control; quantification of the Western blotting signals from OUMS27 (**C**) and JJ012 (**D**) cells with at least 3 independent experiments, (*: *p* < 0.05, **: *p* < 0.01, ***: *p* < 0.001, ^$^: *p* < 0.08). Original images can be found in Appendix A.

**Table 1 biomolecules-14-01071-t001:** Parameters of the survival curves (according to the curve fit, *n* = 3) for OUMS27, JJ012 and MC615 cells following exposure to X-ray or C-ions with and without Olaparib. D10: irradiation dose allowing 10% of surviving fraction; RBE: relative biological effectiveness of C-ions, compared to X-ray, using the D10 values; ER: enhancement ratio of Olaparib, using the D10 values (without/with Olaparib) for each irradiation condition. * = significant differences between with and without Olaparib.

		D10	RBE	ER
OUMS27	X-ray	6.17		
	X-ray + Olaparib	5.18		1.19 *
	C-ions	2.02	3.05	
	C-ions + Olaparib	2.22		0.91
JJ012	X-ray	7.5		
	X-ray + Olaparib	5.04		1.5 *
	C-ions	2.24	3.37	
	C-ions + Olaparib	2.22		1.01
MC615	X-ray	7.58		
	X-ray + Olaparib	5.52		1.37 *
	C-ions	2.36	3.21	
	C-ions + Olaparib	2.27		1.04

**Table 2 biomolecules-14-01071-t002:** Mutation status of JJ012 and OUMS27 cells, with the genes mutated, the type of mutation, the location of the mutation. GIScar score (Genomic Instability Scar) corresponds to a predictive value of genomic instability and HRD status of cell lines, according to [32].

Cell Line	Gene	Event	Mutation	GIScar Score	Number of Large Genomic Events *
JJ012	*IDH1*	missense substitution	p.Arg132Gly		
	*SLX4*	missense substitution	p.Thr345Asn		
	*TP53*	missense substitution	p.Gly199Val		6
	*ARID1A*	frameshift	p.Gln1552Hisfs*21	0.154	
	*ARID1B*	missense substitution	p.Pro1019Leu		
	*ARID1B*	frameshift	p.Gly351Metfs*11		
OUMS27	*TP53*	frameshift	p.Ser149Tyrfs*31	0.408	27
	*EMSY*	intronic mutation	g.76260993C>A		

* Large genomic deletion/duplication with size > 8 Mb.

## Data Availability

Data are contained within the article and Appendix A.

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
