# Peer review of "Radiosensitizing Effect of PARP Inhibition on Chondrosarcoma and Chondrocyte Cells Is Dependent on Radiation LET"

_biomolecules, 2024, doi:10.3390/biom14091071_

Round 1

Reviewer 1 Report

Comments and Suggestions for Authors

In their manuscript „Radiosensitizing effects of PARP inhibition on chondrosarcoma and chondrocyte cells is dependent on radiation LET”, Gilbert and colleagues analyze the effect of the PARP inhibitor Olaparib on the efficacy of Y-ray or C-ion radiotherapy on two chondrosarcoma cell lines (OUMS27 and JJ012). The authors demonstrate an enhanced relative biological effectiveness of C-ion radiotherapy in comparison to X-rays, which is not further increased after PARP inhibition by treatment with 2 µM Olaparib. In contrary, PARP inhibition in combination with X-ray radiotherapy showed an around 20 - 50% reduction of the clonogenic survival of the analyzed chondrosarcoma cell lines in comparison to X-ray radiotherapy alone. Finally, the authors observed an increased expression of the DNA repair related proteins PARP-1, p-ATM, p-RCA-1 and p-DNA-PKc after PARP inhibition in OUMS27 and of the latter two in JJ012. The authors conclude, that PARP inhibition bears potential in combination with X-ray radiotherapy in chondrosarcoma; however, that DNA repair mechanisms and cell responses to DNA damage have a high complexity. The manuscript is written understandable, and the methodology and the data seem sound. However, I have some concerns on this manuscript before recommending publication.

Here are my comments in detail:

1) The discussion section is the weakest part of the manuscript, with some general information (l. 360 - 385) and mainly a repetition of the results of the study (l. 385 - 434) lacking a proper discussion under the existing literature. For instance, in a quick PubMed search I found PMID: 27659937; PMID: 38278469; PMID: 37902934; PMID: 36786286; PMID: 38002066. I would recommend expanding and deepening the discussion and compare the study’s results to the already existing data on the research topic.

2) The quality of the figures should be enhanced. For instance, it is hard to distinguish circles, triangles and squares in Figure 1, and Figure 4 A - C appear to be distorted. Figure 5 C - D: the y-axis titles and ciphers are hardly readable.

3) Table 1: please replace commas by dots as denominators. Table 2: line numbers should be omitted from table.

4) Some minor typos: l.59: agents such as PARP…; l. 62 and 73: some commas too much;  l. 109 and 111: spaces missing; l. 112: in 25 cm2 flasks; l.136: (Covaris); l. 241: Western; l.241: „from Figure 4C“ instead of “in “C””; and others. Please add another proof-reading round.

ü) l. 450 - 469 - still the placeholder text of the template.

Finally, I want to thank the authors for sharing their results with the scientific community. Best regards.

Comments on the Quality of English Language

Some typos and grammatic inconsistencies. Manuscript needs another proof-reading round. 

Author Response

Reviewer 1

In their manuscript „Radiosensitizing effects of PARP inhibition on chondrosarcoma and chondrocyte cells is dependent on radiation LET”, Gilbert and colleagues analyze the effect of the PARP inhibitor Olaparib on the efficacy of Y-ray or C-ion radiotherapy on two chondrosarcoma cell lines (OUMS27 and JJ012). The authors demonstrate an enhanced relative biological effectiveness of C-ion radiotherapy in comparison to X-rays, which is not further increased after PARP inhibition by treatment with 2 µM Olaparib. In contrary, PARP inhibition in combination with X-ray radiotherapy showed an around 20 - 50% reduction of the clonogenic survival of the analyzed chondrosarcoma cell lines in comparison to X-ray radiotherapy alone. Finally, the authors observed an increased expression of the DNA repair related proteins PARP-1, p-ATM, p-BRCA-1 and p-DNA-PKc after PARP inhibition in OUMS27 and of the latter two in JJ012. The authors conclude, that PARP inhibition bears potential in combination with X-ray radiotherapy in chondrosarcoma; however, that DNA repair mechanisms and cell responses to DNA damage have a high complexity. The manuscript is written understandable, and the methodology and the data seem sound. However, I have some concerns on this manuscript before recommending publication.

Here are my comments in detail:

1) The discussion section is the weakest part of the manuscript, with some general information (l. 360 - 385) and mainly a repetition of the results of the study (l. 385 - 434) lacking a proper discussion under the existing literature. For instance, in a quick PubMed search I found PMID: 27659937; PMID: 38278469; PMID: 37902934; PMID: 36786286; PMID: 38002066. I would recommend expanding and deepening the discussion and compare the study’s results to the already existing data on the research topic.

Answer: thank you for this comment. We improved the discussion section accordingly, by adding new references to previous works in the same topic. Several sentences are added to the discussion to enrich our comments

2) The quality of the figures should be enhanced. For instance, it is hard to distinguish circles, triangles and squares in Figure 1, and Figure 4 A - C appear to be distorted. Figure 5 C - D: the y-axis titles and ciphers are hardly readable.

Answer: we improved the quality of figures 1, 4 and 5, as asked

3) Table 1: please replace commas by dots as denominators. Table 2: line numbers should be omitted from table.

Answer: we modified tables 1 and 2 as asked

4) Some minor typos: l.59: agents such as PARP…; l. 62 and 73: some commas too much;  l. 109 and 111: spaces missing; l. 112: in 25 cm2 flasks; l.136: (Covaris); l. 241: Western; l.241: „from Figure 4C“ instead of “in “C””; and others. Please add another proof-reading round.

Answer: thank you for this deep analysis. We modified the points highlighted and checked again the text.

ü) l. 450 - 469 - still the placeholder text of the template.

Answer: we finished this part of the manuscript with author contribution and fundings.

Finally, I want to thank the authors for sharing their results with the scientific community. Best regards.

Answer: thank you once again for your constructive review

Reviewer 2 Report

Comments and Suggestions for Authors

The manuscript by Gilbert and collegues reports an interesting study on the role of PARP inhibition in mediating rediosensitivy in chondrosarcoma cell lines. 

Points to be addressed before publication:

-Figures 1 and 4. The quality of the images should be improved. In particular, the lines corresponding to the different cell lines are difficult to read.

- Figure 5. The bar graphs should be in same order as the WB images.

Author Response

Reviewer 2

The manuscript by Gilbert and collegues reports an interesting study on the role of PARP inhibition in mediating rediosensitivy in chondrosarcoma cell lines.

Points to be addressed before publication:

Comment 1: Figures 1 and 4. The quality of the images should be improved. In particular, the lines corresponding to the different cell lines are difficult to read.

Answer: as asked by reviewer2, we improved the quality of figures 1 and 4

Comment 2: Figure 5. The bar graphs should be in same order as the WB images.

Answer: we changed the order of image in the figure 5 to keep the same order as the WB images

Reviewer 3 Report

Comments and Suggestions for Authors

Dear authors,

Biomolecules-3128162-

Radio sensitizing effects of PARP inhibition on chondrosarcoma and chondrocyte cells is dependent on radiation LET by GILBER et al is an interesting article. The authors evaluated the effects of DNA damage repair inhibitors and high LET irradiation on chondrosarcoma versus chondrocyte cells. The study reveals that LET combination with Olaparib the PARP inhibitor drastically reduced cell survival with there-fold as compared to low LET. Olaparib significantly inhibited PARylation in all the experimental models. A significant reduction of cell survival of both chondrosarcoma and chondrocyte cells was observed following the treatment combining Olaparib and X-ray. PARP inhibition induced an increase of PARP-1 expression and a reduced effect on cell survival of WT IDH chondrosarcoma cells. No radio sensitizing effect was observed in cells exposed to Olaparib paired with high LET irradiation. NHEJ was activated in response to high LET irradiation, neutralizing PARP inhibition effect in both chondrosarcoma cell lines. When high-LET irradiation is not available, PARP inhibition could be used in combination with low-LET irradiation, with significant radio-sensitizing effects on chondrosarcoma cells. Chondrocytes may be affected by the treatment combination too, showing the need to preserve normal tissues from radiation fields when this kind of treatment is suggested.

The study is interesting. However, this needs additional data to improve the quality of the manuscript.

Comments:

1.     Fig 5. Please provide cleaved caspase 3 protein signal in Olaparib+LET

2.     In vivo is needed for at least in one cell line

3.     Please evaluate the toxicity Olaparib+LET or Olaparib or LET in non-cancer and cancer cells by cell viability assay or flow cytometry.

Author Response

Reviewer 3

Radio sensitizing effects of PARP inhibition on chondrosarcoma and chondrocyte cells is dependent on radiation LET by GILBER et al is an interesting article. The authors evaluated the effects of DNA damage repair inhibitors and high LET irradiation on chondrosarcoma versus chondrocyte cells. The study reveals that LET combination with Olaparib the PARP inhibitor drastically reduced cell survival with there-fold as compared to low LET. Olaparib significantly inhibited PARylation in all the experimental models. A significant reduction of cell survival of both chondrosarcoma and chondrocyte cells was observed following the treatment combining Olaparib and X-ray. PARP inhibition induced an increase of PARP-1 expression and a reduced effect on cell survival of WT IDH chondrosarcoma cells. No radio sensitizing effect was observed in cells exposed to Olaparib paired with high LET irradiation. NHEJ was activated in response to high LET irradiation, neutralizing PARP inhibition effect in both chondrosarcoma cell lines. When high-LET irradiation is not available, PARP inhibition could be used in combination with low-LET irradiation, with significant radio-sensitizing effects on chondrosarcoma cells. Chondrocytes may be affected by the treatment combination too, showing the need to preserve normal tissues from radiation fields when this kind of treatment is suggested.

The study is interesting. However, this needs additional data to improve the quality of the manuscript.

Comments:

  1. Fig 5. Please provide cleaved caspase 3 protein signal in Olaparib + LET

Answer: we understand this point of reviewer 3. Cleaved Caspase 3 gives an idea of apoptosis processes following cell treatment. In this manuscript, our goal was not to study cell death or cell death types. Nevertheless, it could be a good idea to know if apoptosis can occur following PARP inhibition associated with irradiation. Indeed, in this manuscript, we analysed PARP signal (entire protein) by western blot, and we did not see any cleavage of PARP following treatment, which showed no activation of apoptosis. As observed in figure 5, PARP signal didn't reduce following treatment, at the opposite, this signal increased in OUMS27 cells or was stable in JJ012 cells.

The sentence following was added within the discussion part.

“Moreover, since no PARP-1 cleavage could be observed in both cell lines, no apoptosis occurred following PARP inhibition associated with irradiations.”

  1. In vivo is needed for at least in one cell line

Answer: we agree with reviewer 3 concerning in vivo experiments. It could give an idea of the impact of the immune system on cell death or survival. Unfortunately, such experiments are not allowed in our facility (GANIL). It is not possible for us to run experiments with animal with Carbon ions in our laboratory. But we keep this idea in mind and we will try in the future to run such experiments on others facilities.

  1. Please evaluate the toxicity Olaparib + LET or Olaparib or LET in non-cancer and cancer cells by cell viability assay or flow cytometry.

Answer: as asked by reviewer 3, we added new results related to cell viability assays. These results are now presented as “supplementary data S1” and discussed in the manuscript text.

Round 2

Reviewer 1 Report

Comments and Suggestions for Authors

In their manuscript „Radiosensitizing effects of PARP inhibition on chondrosarcoma and chondrocyte cells is dependent on radiation LET”, Gilbert and colleagues analyze the effect of the PARP inhibitor Olaparib on the efficacy of Y-ray or C-ion radiotherapy on two chondrosarcoma cell lines (OUMS27 and JJ012). The authors demonstrate an enhanced relative biological effectiveness of C-ion radiotherapy in comparison to X-rays, which is not further increased after PARP inhibition by treatment with 2 µM Olaparib. In contrary, PARP inhibition in combination with X-ray radiotherapy showed an around 20 - 50% reduction of the clonogenic survival of the analyzed chondrosarcoma cell lines in comparison to X-ray radiotherapy alone. Finally, the authors observed an increased expression of the DNA repair related proteins PARP-1, p-ATM, p-RCA-1 and p-DNA-PKc after PARP inhibition in OUMS27 and of the latter two in JJ012. The authors conclude, that PARP inhibition bears potential in combination with X-ray radiotherapy in chondrosarcoma; however, that DNA repair mechanisms and cell responses to DNA damage have a high complexity. This is the first revision of the manuscript.

The authors have responded to all my comments to my satisfaction. I do not have further remarks and recommend publication.

Best regards.

Comments on the Quality of English Language

English is fine - minor spell check might be needed.

Reviewer 3 Report

Comments and Suggestions for Authors

Dear Authors,

The manuscript is improved